# High-Speed Light Signal Transmitters for Optical Communication Based on Ultraviolet Radiation

**Xin Li [1,*], Yue Wu [2], Jialei Yuan [1], Shuyu Ni [1], Chuan Qin [1], Yan Jiang [1], Jie Li [1] and Yongjin Wang [1,*]**

[1] Grünberg Research Centre, Nanjing University of Posts and Telecommunications, Nanjing 210003, China; yuanjialei@njupt.edu.cn (J.Y.); nishuyu@hotmail.com (S.N.); qinchuan1990@hotmail.com (C.Q.); jiangyan@njupt.edu.cn (Y.J.); lijie@njupt.edu.cn (J.L.)

[2] Bell Honors School, Nanjing University of Posts and Telecommunications, Nanjing 210023, China; Q17010202@njupt.edu.cn

[*] Correspondence: lixin1984@njupt.edu.cn (X.L.); wangyj@njupt.edu.cn (Y.W.)



**Featured Application: Optical light communications.**

**Abstract:** A light signal transmitter based on ultraviolet radiation is realized on GaN-on-silicon platform. The light signal transmitter with ultra-small active area is fabricated by a double-etching process. The absolute value of negative junction capacitance of transmitter is as low as the pF (picofarads) scale in positive bias voltage. Small capacitance is beneficial to improve the communication performance of a transmitter. The dominant EL (electroluminescence) peak of transmitter is located at about 380 nm in the ultraviolet range. With the increase of the current, the dominant peak of transmitter remains stable and the light output power is lineally modulated. A free-space data transmission test in the ultraviolet range with 250 Mbps was conducted to indicate a promising high-speed optical communication capability of a light signal transmitter in the ultraviolet range.

**Keywords:** LED (light emitting diode); ultraviolet; GaN; optical communication

## 1. Introduction

UV (ultraviolet) radiation, including the UV-A band (320–400 nm), UV-B band (280–320 nm), and UV-C band (100–280 nm), has extensive applications, such as in light sources, photo catalysts, sewage treatment and communications [1,2]. As the third generation semiconductor material, III-Nitride materials have excellent optoelectronic performance. GaN with a 3.4 eV bandgap can be composed of MQWs (multiple quantum wells) with InN (1.9 eV) and AlN (6.2 eV). The corresponding direct bandgap of MQWs covers the UV range and deep ultraviolet (DUV) range, and optoelectronics devices based on III-Nitrides.

MQWs have been a significant area of research in recent years [3–5]. Optical communication systems transmit information through modulated optical signals from ultraviolet to infrared. The motivation for developing ultraviolet optical communication contains unique channel characteristics, rapid evolution of III-Nitride devices, and emerging application demands in the fields of military and environmental protection and so on [6]. However, due to the limitation of speed and optoelectronic performance of key devices (such as transmitters) based on ultraviolet radiation, the performance of ultraviolet optical communication systems needs further improvement.

For early ultraviolet optical communication systems, flashtubes and lamps with the disadvantages of being bulky, power hungry and with low bandwidth, were selected as light signal transmitters [6].

Consequently, the transmission speed of early ultraviolet optical communication systems was limited around Kbps~Mbps [7,8]. Recently, III-Nitride LEDs with a high efficiency provided a potential application as light signal transmitter for ultraviolet optical communication systems [9–11]. Chang et al. demonstrated that AlN/GaN/InGaN MQWs ultraviolet LEDs obtained high output power at 375/395 nm [12]. Feng et al. reported a successful fabrication of room-temperature electrically injected InGaN/AlGaN MQWs near-ultraviolet laser diode at 390 nm, which was grown on Si by preparing an Al-composition step downgraded AlN/AlGaN multilayer buffer [13].

Our group has studied III-Nitride integrated optoelectronics devices on silicon substrate as a light signal transmitter/receiver for visible and ultraviolet optical communication [14–16]. III-Nitride devices with MQWs have a stable carrier recombination lifetime, and the modulation bandwidth and transmission speed are significantly influenced by RC (resistance-capacitance) time [17]. Reducing the active area of III-Nitride devices with MQWs used in visible and ultraviolet optical communication systems, the modulation bandwidth and transmission speed could be improved by decreasing capacitance [18].

A high-speed light signal transmitter based on ultraviolet radiation is realized on a commercial GaN-on-silicon platform in this paper. The light signal transmitter with ultra-small active area is obtained by a double-etching process. We characterize the optoelectronic properties of the light signal transmitter. The absolute value of the negative junction capacitance of the transmitter is as low as the pF scale in positive bias voltage. The emission peak remains stable at 380 nm in the ultraviolet range under different currents. A free space communication test presents that the light signal transmitter has a high transmission speed in the ultraviolet range. This study provides a potential approach for a light signal transmitter with high speed used in ultraviolet optical communication systems. The schematic of the high-speed light signal transmitter for optical communication based on ultraviolet radiation is shown in Figure 1.

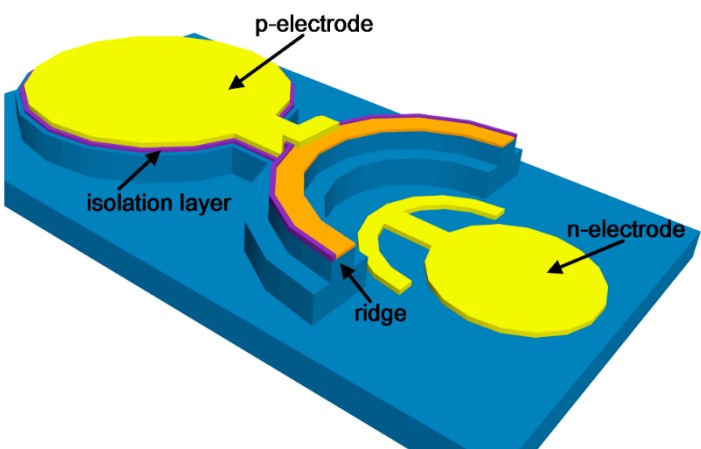

**Figure 1.** Schematic of high-speed light signal transmitter for optical communication based on ultraviolet radiation.

## 2. Materials and Methods

III-Nitride epitaxial layer of GaN-on-silicon platform (Lattice Power Corporation, Suzhou, China) consists of ~25 nm thick p-contact GaN layer, ~500 nm thick p-cladding AlGaN layer, ~52 nm InGaN/GaN MQWs (4 pairs of quantum well, 3 nm InGaN/10 nm $Al_{0.1}Ga_{0.9}N$ per pair), ~750 nm thick n-cladding AlGaN layer, ~2.45-μm-thick n-contact AlGaN layer, ~1030 nm thick undoped GaN layer and ~750 nm thick AlGaN buffer layer. Metal films with Pd 30 nm/Pt 30 nm/Au 50 nm are deposited on III-Nitride epitaxial layer by electron beam (EB) evaporation (step a). The structures of the active area are patterned on metal films and transferred to III-Nitride epitaxial layer by inductively coupled plasma (ICP) etching for III-V materials with a 320 nm etching depth (step b). Then, a silicon oxide

layer for electrical isolation is deposited on III-Nitride epitaxial layer by plasma enhanced chemical vapor deposition (PECVD), and it is patterned by photolithography and wet etching with a BHF (buffer hydrofluoric acid) etching solution (step c). The region for exposing n-type III-nitride is patterned and etched by ICP etching with 2 μm depth (step d). Finally, p/n electrodes (Ti 50 nm/Pt 100 nm/Au 500 nm) are formed by EB evaporation and lift-off process (step e). Figure 2 shows the fabrication process of the light signal transmitter based on ultraviolet radiation.

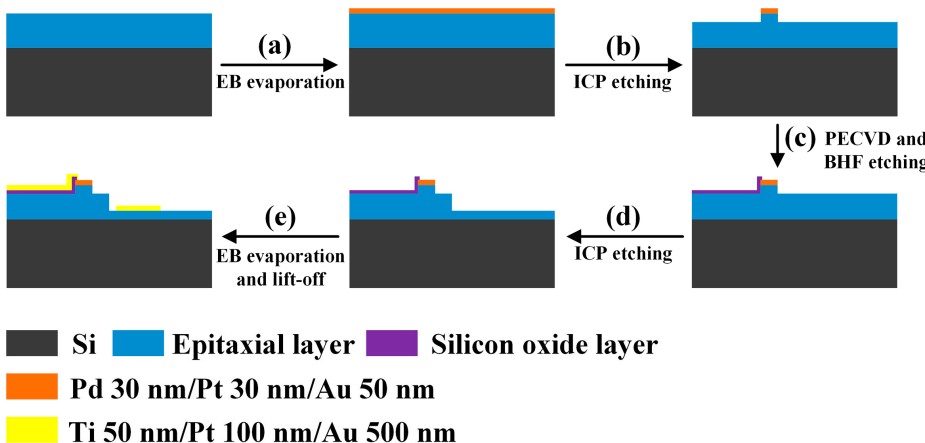

**Figure 2.** Fabrication process of the light signal transmitter based on ultraviolet radiation.

Figure 3 shows the optical microscope image of the light signal transmitter based on ultraviolet radiation. The inset image in Figure 3 is an enlarged SEM (scanning electron microscope) image of the active area of the light signal transmitter. As mentioned above, the modulation bandwidth and transmission speed of the ultraviolet optical communication systems are significantly influenced by the absolute value of transmitter capacitance, which depends on the area of the active area. The light signal transmitter with an ultra-small active area is fabricated by a double-etching process.

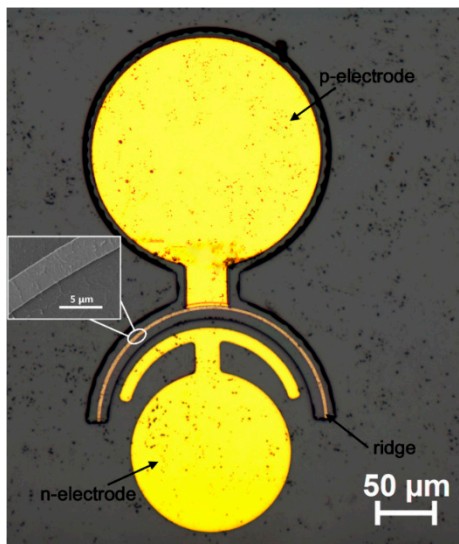

**Figure 3.** Optical microscope image of the light signal transmitter based on ultraviolet radiation. Inset: Enlarged SEM (scanning electron microscope) image of the active area (ridge structure) of the light signal transmitter.

Figure 4a presents three-dimensional AFM (atomic force microscope) images of the active area (ridge structure) of the light signal transmitter. The roughness of the top surface of the ridge structure

with electrode Pd/Pt/Au is less than 50 nm. As shown in Figure 4b, the width and height of the active area is about 2.74 μm and 365 nm. The sidewall of the active area has high steepness and small roughness. The thickness of Pd/Pt/Au electrode is 110 nm, and the etching depth of the active area is about 255 nm. P-type GaN layer and part of p-cladding AlGaN layer are removed by the first etching step. The ultra-small active area has a width of 2.74 μm and an area of 850 μm$^2$. The absolute value of transmitter capacitance could be greatly reduced. The etching depth of the second etching step for exposing n-type III-Nitride is 2.3 μm, etching to n-contact AlGaN layer. The deep etching for n-type III-Nitride could reduce parasitic resistance and capacitance, thus the second etching step is also beneficial to improve the modulation bandwidth and transmission speed of the transmitter [19,20].

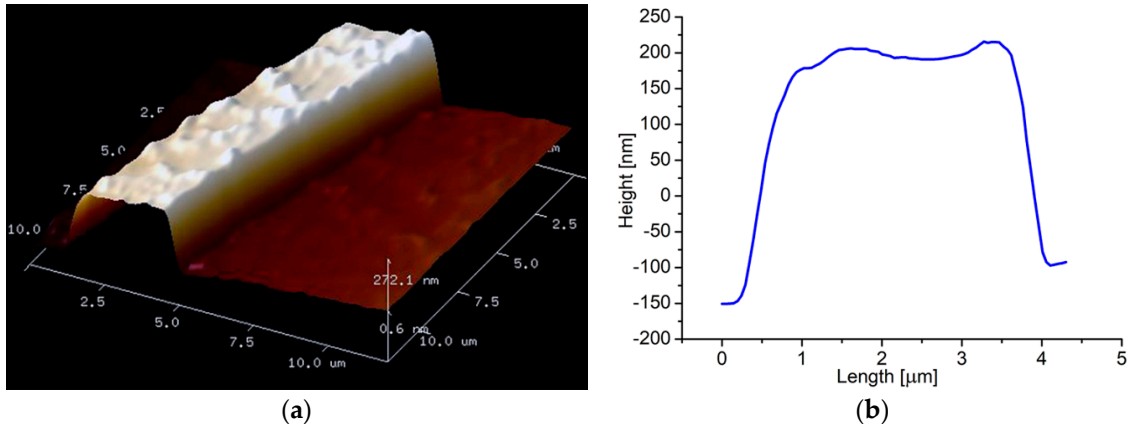

(a)　　　　　　　　　　　　　　　　　　　　　　　(b)

**Figure 4.** (**a**) Three dimensional AFM (atomic force microscope) images of the active area (ridge structure) of the light signal transmitter; (**b**) cross-section size of the active area.

## 3. Results and Discussion

Electroluminescence (EL) and light output power-current-voltage (L-I-V) characteristics of the light signal transmitter are measured by a probe station connected to a semiconductor device parameter analyzer (Agilent, B1500A, Santa Clara, CA, USA), a spectrometer in the ultraviolet range and an optical power meter. The dominant peak of the EL spectrum is about 380 nm in the ultraviolet range as shown in Figure 5a. The transmitter exhibits typical current-voltage characteristics (black curve in Figure 5b) of a LED, with a relatively low turn-on voltage (about 3 V). Light output power versus the current measured by microscopy system and fibred-coupled optical power meter is a linear growth curve (blue curve in Figure 5b). With the increase of current, the dominant peak of the transmitter remains stable and the light output power is lineally modulated. It demonstrates that the light signal transmitter has a suitable optoelectronics performance for optical communication in the ultraviolet range.

As can be seen in Figure 6, C-V (current-voltage) characteristics of the light signal transmitter presents the negative capacitance behavior under AC signal with different frequency (10 KHz–1 MHz) in positive bias voltage range. The junction capacitance is positive under negative bias voltage. As the positive bias voltage increases, the junction capacitance decreases and drops down to negative value after reaching the turn-on voltage in positive bias voltage range. The modulation bandwidth and transmission speed could be improved by reducing RC time constant, and a lower RC time constant is always accompanied with lower absolute value of negative junction capacitance under positive bias voltage [20,21]. The absolute value of negative junction capacitance increases with lower AC signal frequency and higher positive bias voltage. Under 4 V bias voltage, the absolute value of negative junction capacitance with 10 KHz, 100 KHz and 1 MHz AC signal are 105.3 pF, 58.3 pF and 22.5 pF, respectively. Because there are few studies about the capacitance characterization of a light emitter in the ultraviolet range, we compared the absolute value of negative junction capacitance in our study with the capacitance characterization of a III-Nitrides light emitter in the visible range [22–24].

It demonstrates that the absolute value of the negative junction capacitance in our study is smaller than the III-Nitrides light emitter in the visible range in the above references.

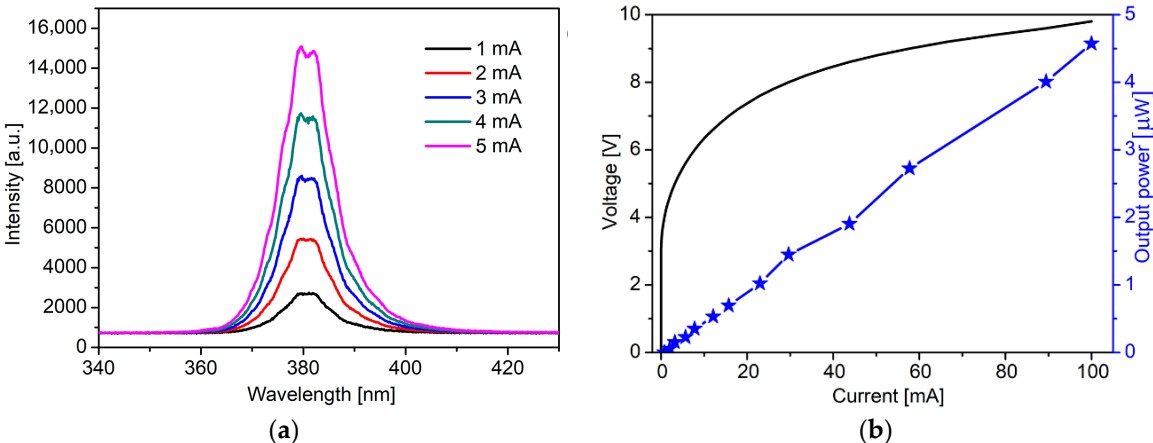

**Figure 5.** (**a**) Electroluminescence (EL) characteristics verse current of the light signal transmitter; (**b**) light output power-current-voltage (L-I-V) characteristics of the light signal transmitter.

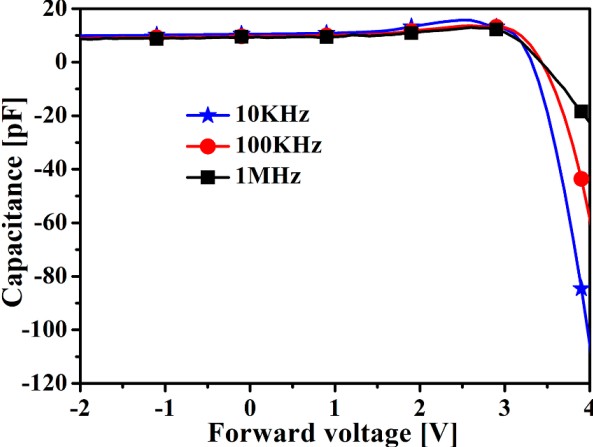

**Figure 6.** C-V (current-voltage) characteristics of the light signal transmitter under AC signal with different frequency.

A free-space data transmission test as shown in Figure 7a was conducted to study the transmission speed of the light signal transmitter. The transmit signal was loaded on light signal transmitter by arbitrary waveform generator (Keysight, 81160A, Santa Rosa, PH, USA) with 250 Mbps random binary sequence. A 40× objective lens with a numerical aperture of 0.75 was set 80 cm from the transmitter to capture optical signal. The optical signal was sent to avalanche photodiode (Hamamatsu, C12702–12, Hamamatsu, Japan) to amplify the received light signal to electrical signal. The electrical signal was divided into two channels with low-pass and high-pass filters, and is finally characterized by an Agilent DSO9254A digital storage oscilloscope. The transmit signal and received signal are shown in Figure 7b, and the signal waveform is well preserved during transmission. The clear open eye diagrams of free-space data transmission test at 250 Mbps are clearly observed in Figure 7c. Three-dB modulation bandwidth of transmitter at a 40 mA current is shown in Figure 7d, and it achieves about 260 MHz (tested by vector network analyzer, Keysight, E5080A). It demonstrates that the light signal transmitter is capable of achieving high-speed transmission in optical communication in the ultraviolet range. There is still great potential for the improvement of the modulation bandwidth and transmission speed of the light signal transmitter in the ultraviolet range.

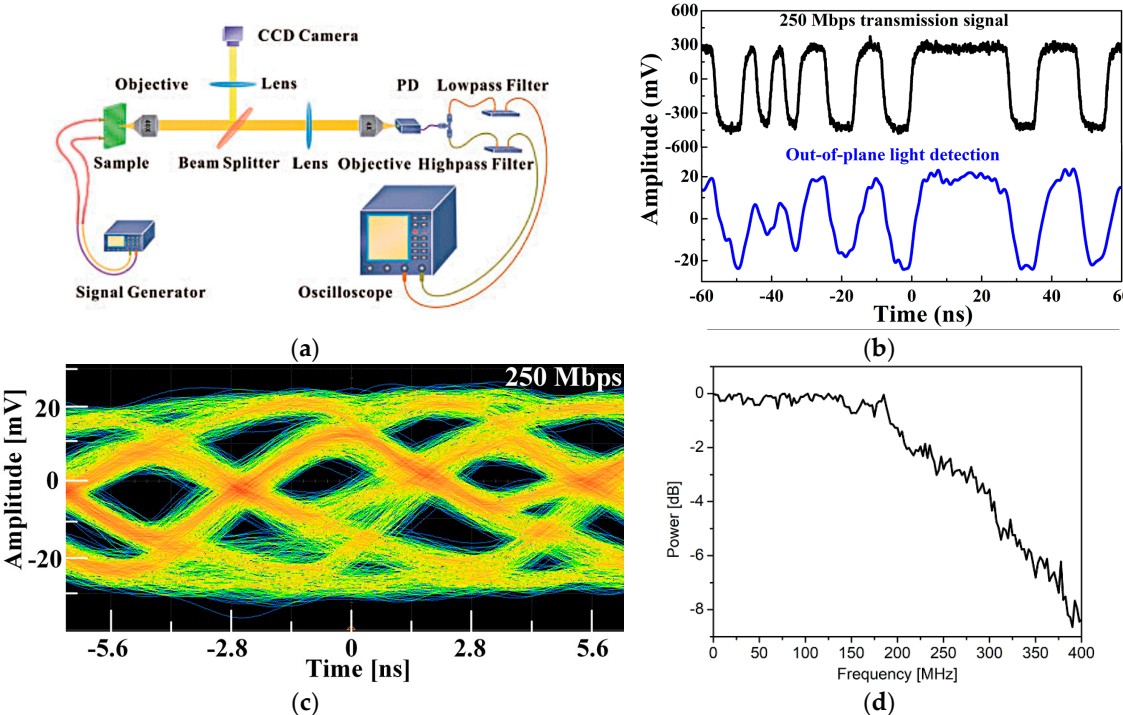

**Figure 7.** (**a**) Free-space data transmission test setup; (**b**) transmit signal loaded on light signal transmitter with 250 Mbps random binary sequence and received signal captured by avalanche photodiode module; (**c**) eye diagrams of the light signal transmitter measured at 250 Mbps; (**d**) 3 dB modulation bandwidth of transmitter at 40 mA current.

## 4. Conclusions

A high-speed light signal transmitter based on ultraviolet radiation is realized on GaN-on-silicon platform. The modulation bandwidth and transmission speed of the light signal transmitter is significantly influenced by the absolute value of negative junction capacitance in positive bias voltage range, which depends on the size of the active area. We fabricated the light signal transmitter with ultra-small active area (850 $\mu m^2$) by a double-etching process. The dominant EL peak of transmitter is about 380 nm in the ultraviolet range. With the increase of current, the dominant EL peak remains stable and the light output power is lineally modulated. Under 4 V bias voltage, the absolute value of negative junction capacitance of the transmitter with 10 KHz, 100 KHz and 1 MHz AC signals are 105.3 pF, 58.3 pF and 22.5 pF, respectively. A free-space data transmission test in the ultraviolet range with 250 Mbps was conducted to present that the high-speed optical communication in the ultraviolet range is feasible for a light signal transmitter in our study.

**Author Contributions:** Conceptualization, X.L. and Y.W. (Yongjin Wang); methodology, X.L. and Y.W. (Yue Wu); data curation, J.Y., S.N. and Y.J.; investigation, X.L., Y.W. (Yue Wu) and C.Q.; writing—original draft preparation, X.L.; writing—review and editing, J.L.; visualization, X.L.; supervision, X.L.; project administration, Y.W. (Yongjin Wang). All authors have read and agreed to the published version of the manuscript.

**Funding:** This research was funded by China Postdoctoral Science Foundation funded project (2018M640508); Natural Science Foundation of the Jiangsu Higher Education Institutions (18KJB510025); National Natural Science Foundation of China (61322112, 61531166004 and 11847022); 1311 Talent Program of Nanjing University of Posts and Telecommunications, National Self-Funding Project of Nanjing University of Posts and Telecommunications (NY218013); Special Project for Intergovernment Collaboration of State Key Research and Development Program (2016YFE0118400); Natural Science Foundation of Jiangsu Province (BE2016186 and BK20170909).

**Acknowledgments:** Thanks to the Lattice Power Corporation for providing GaN-on-silicon platform.

**Conflicts of Interest:** The authors declare no conflict of interest.

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
