# Peer review of "High-Speed Light Signal Transmitters for Optical Communication Based on Ultraviolet Radiation"

_applsci, doi:10.3390/app10020693_

Round 1
Reviewer 1 Report
Author fabricated an ultraviolet light signal transmitter based on GaN-silicon platform. Achieved active area is 850 µm2. EL spectra is observed and capacitance is also measured with bias voltage. Finally author showed free space data transmission at 250 mbps speed.
This work is useful for designing next generation transmitter. I have the following questions before I make any recommendation-
Author provide some background information, however, since the main claim of author is about ultra small active area, I would like to know what is the minimum active area realized in the literature and what is the speed? Why the absolute value of negative junction capacitance increases with lower AC signal frequency and higher positive bias voltage. During free space test Author set the objective lens at 80 cm away from the light signal transmitter to capture optical signal. How the transmitted signal will depend on this distance?Author Response
Point 1: Author provide some background information, however, since the main claim of author is about ultra-small active area, I would like to know what is the minimum active area realized in the literature and what is the speed?
Response 1: The minimum active area realized in the literature is a square with 40 μm length (1600 μm2) in literature [20] Yang, W.; Zhang, S.; McKendry, J. J.; Herrnsdorf, J.; Tian, P.; Gong, Z.; Ji, Q.; Watson, I. M.; Gu, E.; Dawson, M. D.; Feng, L.; Wang, C.; Hu, X. Size-dependent capacitance study on InGaN-based micro-light-emitting diodes. J. Appl. Phys. 2014, 116, 044512, doi: 10.1063/1.4891233. Literature [20] focuses on the electrical performance of LED, and there is no communication test.
The highest communication speed in all the literature is 540MHz 3dB modulation bandwidth in literature [19] Rashidi, A.; Monavarian, M.; Aragon, A.; Okur, S.; Nami, M.; Rishinaramangalam, A.; Mishkat-Ul-Masabih, S.; Feezell, D. High-speed nonpolar InGaN/GaN LEDs for visible-light communication. IEEE Photonics Technol. Lett. 2017, 29, 381-384, doi: 10.1109/LPT.2017.2650681. The active area in literature [19] is a circle with 60 μm diameter (2826 μm2). The communication test in literature [19] is in visible range, not in ultraviolet range.
In all the literature, the highest communication speed in ultraviolet range is 5MHz 3dB modulation bandwidth in literature [8] Xing, Y.; Zhang, M.; Han, D.; Ghassemlooy, Z. Experimental study of a 2× 2 MIMO scheme for ultraviolet communications, In Proceedings of 15th International Conference on Optical Communications and Networks (ICOCN), Hangzhou, China, 24-27 September 2016.
Point 2: Why the absolute value of negative junction capacitance increases with lower AC signal frequency and higher positive bias voltage.
Response 2: When the AC signal frequency increases, the carriers from sub-band gap defects cannot follow the change of AC signal. Then the carriers will be trapped at the sub-band gap due to finite inertia. According to Bansal-Datta-model, the carrier consumption at higher AC signal frequency will reduce the absolute value of negative junction capacitance. The explanation about relationship between absolute value of negative junction capacitance and AC signal frequency is shown in reference [20] (Page 5, Paragraph 4). The details about Bansal-Datta-model is shown in “Bansal, Kanika, and Shouvik Datta. Voltage modulated electro-luminescence spectroscopy to understand negative capacitance and the role of sub-bandgap states in light emitting devices. Journal of Applied Physics 110.11 (2011): 114509.”
When the positive bias voltage increases, a lot of carriers will pass through the depletion region and the depletion approximation is no longer applicable. In this case, the carriers that are stored in either the n-type or p-type diffusion region change with the applied bias and this capacitance
The dependence of negative capacitance on voltage is shown in reference [23] (Page 2, Paragraph 9) as below.
where Cp is the value of negative capacitance, C0 is a constant for a given LED. At room temperature m is a factor. V is the value of positive bias voltage. According to the above equation, the absolute value of negative junction capacitance increases with higher positive bias voltage.
Point 3: During free space test Author set the objective lens at 80 cm away from the light signal transmitter to capture optical signal. How the transmitted signal will depend on this distance?
Response 3: The divergence and loss of transmitted signal will occur when emitted light of transmitter travels in free space. The greater the distance between transmitter and avalanche photodiode in free space, the weaker the transmitted light intensity. The focal length of objective lens, that is, the distance between objective lens and avalanche photodiode is fixed. Therefore, the distance between transmitter and avalanche photodiode mainly depends on the distance between the objective lens and transmitter.
In free space test, the transmitted signal is captured by avalanche photodiode (Hamamatsu, C12702–12) to amplify the optical signal to electrical signal. Due to the limited sensitivity of avalanche photodiode, and the intensity of emitted light is not too strong, it is difficult for avalanche photodiode to detect the transmitted signal if the distance between transmitter and avalanche photodiode was too large. By adjusting between the objective lens and transmitter in the test, we found that the avalanche photodiode we used in free space test is very difficult to detect the signal when the distance was greater than 80cm.
Reviewer 2 Report
The manuscript reports on the manufacture and basic testing of the ultraviolet light signal transmitter. The presented information is interesting and it contributes to the progress of UV optoelectronics. The manuscript should be published in Applied Sciences after authors amend few minor items listed below.
1. Authors should briefly explain the construction, dimensions and shape of contacts in their device. Why just this arrangement is convenient for the device? Are given dimensions deduced from theoretical simulations or obtained by the optimization procedure?
2. The electroluminescence shown in Fig. 5(a) reveals weak splitting of the main peak. Authors could comment this feature.
Typing and grammar faults:
line 27: The abbreviation 'MQW' should be defined. AIN -> AlN
line 65: p-contact
lines 65-67: The composition of ternary materials should be defined.
line 66: '4 pairs of MQW' - Authors perhaps mean SQW, single quantum well.
line 103: 'range'
line 116: 'positive'
line 117: 'improved'
line 123: improper grammar
line 136 'signals are divided'
Author Response
Point 1: Authors should briefly explain the construction, dimensions and shape of contacts in their device. Why just this arrangement is convenient for the device? Are given dimensions deduced from theoretical simulations or obtained by the optimization procedure?
Response 1: At present, the setting of the construction, dimensions and shape of contacts mainly depends on process experience and equipment conditions in this paper. In order to minimize the capacitance of transmitter, we need to reduce active area as much as possible. The active area of transmitter in our study depends on the dimensions of p-contact on ridge structure. The width of p-contacts on ridge structure is the same as that of active area, which is 2.74 μm. The limit of line width of lithography machine in our lab is 2 μm. Actually, we originally designed two kinds of transmitter with 2 μm and 3 μm width p-contact on ridge structure. After etching, the fabrication of transmitter with 2 μm width p-contact on ridge structure was poor. Therefore, we presented the transmitter with design value of 3 μm width. After etching, the actual width is reduced to 2.74 μm.
p electrode and n electrode are designed as arc structures close to each other to ensure good current distribution based on our optimization experience. The arc structure has no obvious sharp angle and curvature change, which is beneficial to the uniform distribution of current on the contacts. We are recently learning the electrostatic module of COMSOL simulation software. We will perform simulation of current distribution under different contacts in future studies according to the comment of reviewer to improve the design of contacts.
The circular parts of p/n contacts have 140 μm and 200 μm diameter respectively. When testing the transmitter on the probe table, the circular parts are mainly used to contact the probe. The relatively large size of circular parts could ensure that p/n contacts are easy to maintain full and stable contact with probe.
Point 2: The electroluminescence shown in Fig. 5(a) reveals weak splitting of the main peak. Authors could comment this feature.
Response 2: We assume that the weak splitting of the electroluminescence main peak is due to the interference phenomenon of emitted light. Emitted light is reflected many times in the upper and lower interface of thick GaN-based epitaxial layer. The emitted light according with certain wavelength and optical path interferes with each other. The interference phenomenon results in the interference phase length (intensity increase) or interference cancellation (intensity decrease). This phenomenon is more obvious at the electroluminescence main peak with strongest light intensity. In the previous research work in our lab “Qin, Chuan, et al. "Transferrable monolithic multicomponent system for near-ultraviolet optoelectronics." Applied Physics Express 11.5 (2018): 051201”, we used similar GaN-on-silicon platform (Lattice Power Corporation, China) in ultraviolet range. The silicon substrate was removed, and the GaN-based epitaxial layer was thinned from backside in this reference. The number of interference modes of emitted light was reduced. The interference phenomenon was suppressed. The electroluminescence main peak was smooth and no splitting was observed.
Point 3: line 27: The abbreviation 'MQW' should be defined. AIN -> AlN
line 65: p-contact
lines 65-67: The composition of ternary materials should be defined.
line 66: '4 pairs of MQW' - Authors perhaps mean SQW, single quantum well.
line 103: 'range'
line 116: 'positive'
line 117: 'improved'
line 123: improper grammar
line 136 'signals are divided'
Response 3: We corrected these grammatical errors throughout the paper as your comment. (in red)
Round 2
Reviewer 1 Report
English language and style should be improved
Author Response
We corrected some grammatical errors throughout the paper, and improved English language and style within our current capabilities as your comment. Since we are not native English speakers, our English proficiency is currently quite limited. We will try our best to improve English research writing in future. Thank you very much for your comments and suggestions. All modification about English language and style are marked in red in the paper.